# Enhancing Metabolic Efficiency through Optimizing Metabolizable Protein Profile in a Time Progressive Manner with Weaned Goats as a Model: Involvement of Gut Microbiota

Jian Wu,[a,b] Xiaoli Zhang,[a,b] Min Wang,[a] Chuanshe Zhou,[a] Jinzhen Jiao,[a] Zhiliang Tan[a]

[a]CAS Key Laboratory of Agroecological Processes in Subtropical Region, National Engineering Laboratory for Pollution Control and Waste Utilization in Livestock and Poultry Production, Hunan Provincial Key Laboratory of Animal Nutritional Physiology and Metabolic Process, Institute of Subtropical Agriculture, The Chinese Academy of Sciences, Changsha, Hunan, People's Republic of China

[b]University of Chinese Academy of Sciences, Beijing, China

**ABSTRACT** Feeding a growing global population and lowering environmental pollution are the two biggest challenges facing ruminant livestock. Considering the significance of nitrogen metabolism in these challenges, a dietary intervention regarding metabolizable protein profiles with different rumen-undegradable protein (RUP) ratios (high RUP [HRUP] versus low RUP [LRUP]) was conducted in young ruminants with weaned goats as a model. Fecal samples were collected longitudinally for nine consecutive weeks to dissect the timing and duration of intervention, as well as its mechanism of action involving the gut microbiota. Results showed that at least 6 weeks of intervention were needed to distinguish the beneficial effects of HRUP, and HRUP intervention improved the metabolic efficiency of goats as evidenced by enhanced growth performance and nutrient-apparent digestibility at week 6 and week 8 after weaning. Integrated analysis of bacterial diversity, metabolites, and inferred function indicated that HRUP intervention promoted *Eubacterium* abundance, several pathways related to bacterial chemotaxis pathway, ABC transporters, and butanoate metabolism and thereafter elicited a shift from acetate production toward butyrate and branched-chain amino acid (BCAA) production. Meanwhile, three distinct phases of microbial progression were noted irrespective of dietary treatments, including the enrichment of fiber-degrading *Ruminococcus*, the enhancement of microbial cell motility, and the shift of fermentation type as weaned goats aged. The current report provides novel insights into early-life diet-microbiota axis triggered by metabolic protein intervention and puts high emphasis on the time window and duration of dietary intervention in modulating lifelong performance of ruminants.

**IMPORTANCE** Precise dietary intervention in early-life gastrointestinal microbiota has significant implications in the long-life productivity and health of young ruminants, as well as in lowering their environmental footprint. Here, using weaned goats as a model, we report that animals adapted to high rumen-undegradable protein diet in a dynamic manner by enriching fecal community that could effectively move toward and scavenge nutrients such as glucose and amino acids and, thereafter, elicit butyrate and BCAA production. Meanwhile, the three dynamic assembly trajectories in fecal microbiota highlight the importance of taking microbiota dynamics into account. Our findings systematically reported when, which, and how the fecal microbiome responded to metabolizable protein profile intervention in young ruminants and laid a foundation for improving the productivity and health of livestock due to the host-microbiota interplay.

**KEYWORDS** fecal microbiota, metabolizable protein profile, microbial progression, weaned goats, metabolic efficiency

Address correspondence to Jinzhen Jiao, jjz@isa.ac.cn, or Zhiliang Tan, zltan@isa.ac.cn.

The authors declare no conflict of interest.

With the world population reaching 9.2 billion by 2050, there is an urgent demand for sustainable livestock to minimize the competition for resources such as land, water, and cereal grains between human and livestock (1, 2). Pressures from the governments and consumers force the livestock industry to reduce its environmental footprint, including excretion of various pollutants such as methane, ammonia, and nitrous oxide (1, 3). The ruminants are distinguishable from monogastric animals due to their capacity to convert fibrous plants into high-quality animal proteins such as milk and meat (1). If we can improve the animal performance of ruminants, we might be able to not only produce more food with fewer resources but also reduce the negative environmental effects (3, 4).

Ruminants maintain a diverse and complex commensal prokaryotic and eukaryotic microorganism in the gastrointestine, which empowers animals with the ability to convert plant polysaccharides and nonprotein nitrogen into short-chain fatty acids and microbial protein to obtain energy and amino acids (2). Mounting evidence suggests that colonization of gastrointestinal microbiota occurs soon after birth and its assembly takes place in distinct steps during animal development (5–7). Newborn ruminants harbor a facultative anaerobic *Proteobacteria*-dominated bacteria in the gastrointestine, which is gradually replaced by anaerobic *Bacteroidetes* and *Firmicutes* phyla in mature animals (6). Numerous nutritional intervention efforts to shift the microbial fermentation toward more efficient metabolic pathways in order to enhance animal performance in mature ruminants failed due to the high redundancy (overlap of function among multiple species) and resilience (resistance to and capacity to recover from perturbation) of the microbial ecosystem (8, 9). From an ecological perspective, due to its lower diversity and colonization resistance, the gastrointestinal microbial ecosystem of young ruminants is more receptive to dietary intervention than that of adult ruminants (10). As such, this offers a potential to perform dietary intervention in early-life gastrointestinal microbiota to produce long-term impacts into adult life (11). Weaning represents one of the most challenging periods to ruminants in early life, during which the animals gradually shift from a milk-based diet that is digested primarily in the abomasum to a solid feed diet that is digested through ruminal microbial fermentation (12, 13). The drastic shifts in dietary components will cause subsequent weaning stress featured by decreased growth rates and immune disorders (14). Of note, during this transformation, the rumen and lower gut of ruminants face the challenge of protecting the host from pathogens while supporting the metabolism of nutrients for growth. Recent reports in ruminal and fecal microbiome have indicated that compositions of gastrointestinal microbiota are easily altered in response to weaning stress, and further correlation analysis demonstrates their significance in modulating weight gain and feed intake of dairy calves (15). Despite this, there is a paucity of knowledge of how gastrointestinal microbiota develops gradually after weaning and how it responds to dietary intervention.

Nitrogen consists of an essential nutrient component critical for the growth and productivity of ruminants, and its improper excretion contributes to environmental pollution (3, 16). The complex rumen ecosystem makes the nitrogen metabolism more mysterious and challenging than its monogastric counterparts (16). Generally, the metabolizable protein (MP) for ruminants is supplied by combinations of rumen-degradable protein (RDP) and rumen-undegradable protein (RUP) (17). The RUP and microbial protein (MCP) generated by ruminal microbes consist of the amino acids in the intestine, and optimizing metabolizable protein profile with RUP/RDP ratios could potentially improve nitrogen efficiency and decrease environmental pollution in adult ruminants (17–19). However, whether this manipulation is effective in young ruminants is not well documented. To address this knowledge gap, using weaned goats as a model, we conducted a dietary intervention study with different RUP/RDP ratios and speculated that balancing its ratio could optimize metabolic efficiency through modulating gastrointestinal microbiome. Fecal samples were used as a proxy for gastrointestinal microbiota owing to their noninvasive collection nature and suitability for frequent

sampling and because they provide opportunities for large-scale and longitudinal studies (20).

## RESULTS

**HRUP improved the metabolic efficiency as evidenced by enhanced growth performance and apparent nutrient digestibility in weaned goats.** The HRUP treatment dramatically improved ($P < 0.01$) body weight by 24.3%, 22.4%, 23.9%, and 25.3%, respectively, within 5, 6, 7, and 8 weeks after weaning (Fig. 1B). Compared to LRUP, the body length gain increased sharply up to 26.4% in HRUP ($P = 0.033$), while chest grith gain and body height gain remained unchanged ($P > 0.05$) (Fig. 1D). Although a similar feed intake was observed between two treatments during the whole experimental period ($P > 0.10$) (Fig. 1C), evaluation of nutrient-apparent digestibility demonstrated that the digestibility of dry matter (DM), neutral detergent fiber (NDF), crude protein (CP), crude fiber (CF), and ether extract (EE) of the HRUP group during weeks 6 and 8 was greater than that in the LRUP group ($P < 0.05$) (Fig. 1E). In contrast, HRUP treatment did not affect ($P = 0.317$) the ADF digestibility in week 6 but significantly affected ($P = 0.009$) that in week 8. These data indicated that HRUP treatment could improve the metabolic efficiency of weaned goats.

**HRUP had small but significant impacts on serum free amino acid profiles in weaned goats.** Serum free amino acids represent vital components of systemic protein metabolism, and their profile has been reported to be linked to animal performance. As presented in Table 1, serum concentrations of lysine and histidine were increased by 31% ($P = 0.045$) and 33% ($P = 0.002$) in goats fed HRUP diet compared to their counterparts fed LRUP diet. Likewise, valine, leucine, tyrosine, and phenylalanine apparently accumulated more ($P < 0.05$) in the serum of HRUP goats. However, other free amino acids showed similar concentrations ($P > 0.05$) between two treatments.

**Shifts in fecal short-chain fatty acid and branched-chain amino acid profiles were driven mainly by time but slightly affected by HRUP.** Short-chain fatty acids, produced during microbial fermentation of organic matter in the gastrointestinal tract, could exert a wide range of physiological effects on the host. Intriguingly, despite that no interaction ($P > 0.05$) was noted for each individual short-chain fatty acid (SCFA) between week and treatment, HRUP affected the SCFA profile to a lesser extent, as it drastically elevated the molar percentage of butyrate in week 8 ($P < 0.05$). Furthermore, as time increased, the concentration of total SCFA gradually raised from 36.9 mM (LRUP) or 41.9 mM (HRUP) up to 84.6 mM (LRUP) or 87.9 mM (HRUP) in week 8 ($P < 0.01$) after weaning (Fig. 2A). The SCFA profile remained stable during weeks 0 to 5, whereas it shifted toward propionate and butyrate fermentation afterwards, as acetate molar percentage decreased remarkably ($P < 0.01$), while molar percentages of propionate and butyrate increased by 40% and 200% ($P < 0.01$) after week 6 of weaning. With respect to branched-chained fatty acids (BCFAs; isobutyrate, valerate, and isovalerate), their concentrations were quite low (<1.0 mM) during the whole period and fluctuated slightly with time.

Branched-chain amino acids (leucine, isoleucine, and valine), essential amino acids synthesized by gut microbiota, are demonstrated to play critical roles in maintaining homeostasis in mammals by regulating protein synthesis, lipid metabolism, insulin resistance, and immunity. Evidence from our data indicated that their values in feces fluctuated with week, with a wave base appearing at week 5 in both HRUP and LRUP kids (Fig. 2B). Moreover, HRUP treatment drastically elevated ($P < 0.01$) fecal BCAA concentrations during weeks 6, 7, and 8. This coincided with the above-mentioned enhanced valine and leucine levels in the serum.

**Total bacteria copy number and microbial crude protein level of fecal microbiome fluctuated with time while they were unaffected by HRUP treatment.** Fecal samples were taken every week after weaning from goats to profile the microbiome. Real-time PCR analysis indicated that no significant interaction was noted between treatment and week ($P = 0.784$) for the mean absolute number of the overall bacterial 16S rRNA genes (Fig. 3A), and its value remained similar in HRUP and LRUP treatments ($P = 0.136$). Minor

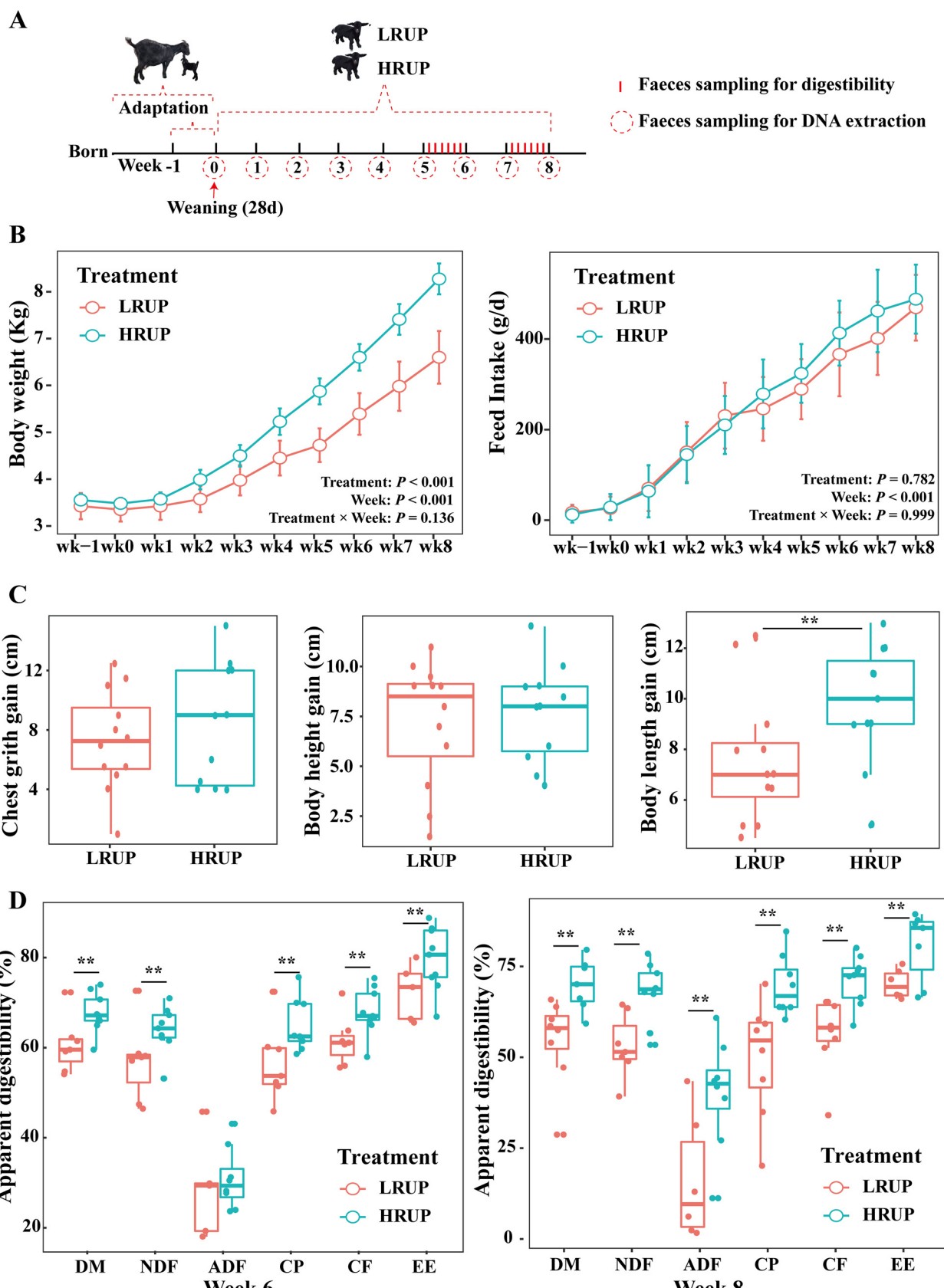

**FIG 1** High rumen-undegradable protein diet (HRUP) remarkably improved the performance of goats. (A) Overview of the study design and sample collection. (B) Body weight was monitored every week during dietary intervention. (C) Feed intake was recorded every week during dietary intervention. (D) Effect of dietary intervention on the gain of chest grith, body height, and body length at the end of experiment. HRUP significantly improved the apparent digestibility of DM, NDF, CP, CF, and EE at 6 and 8 weeks after weaning. **, $P < 0.05$.

**TABLE 1** HRUP affects free amino acid profile in serum ($\mu$g/mL)

| Serum free amino acids | Amino acid profile ($\mu$g/mL) with treatment: | | SEM[a] | P value |
|---|---|---|---|---|
| | LRUP | HRUP | | |
| Aspartic acid | 3.10 | 3.03 | 0.261 | 0.897 |
| Glutamic acid | 12.24 | 12.19 | 0.707 | 0.975 |
| Lysine | 15.94 | 20.86 | 1.213 | 0.045 |
| Histidine | 6.53 | 8.68 | 0.361 | 0.002 |
| Arginine | 26.85 | 31.04 | 1.841 | 0.278 |
| Threonine | 6.94 | 8.47 | 0.430 | 0.085 |
| Serine | 7.50 | 8.21 | 0.341 | 0.325 |
| Glycine | 53.76 | 50.21 | 1.667 | 0.311 |
| Alanine | 20.39 | 21.30 | 1.052 | 0.684 |
| Valine | 22.70 | 29.74 | 1.358 | 0.008 |
| Methionine | 3.95 | 4.51 | 0.196 | 0.169 |
| Isoleucine | 7.85 | 7.97 | 0.373 | 0.879 |
| Leucine | 16.75 | 20.07 | 0.778 | 0.034 |
| Tyrosine | 9.47 | 11.16 | 0.404 | 0.039 |
| Phenylalanine | 7.07 | 8.69 | 0.318 | 0.009 |
| Proline | 8.95 | 9.56 | 0.680 | 0.674 |

[a]SEM, standard error of the mean.

fluctuations were observed among weeks, as absolute number of the overall bacterial 16S rRNA genes amounted to 10.39 $\log_{10}$ 16S rRNA gene copies per gram feces at week 6 compared to the lower numbers of 9.63 $\log_{10}$ 16S rRNA gene copies per gram feces at week 1 ($P = 0.013$). Similarly, there was no significant treatment-week interaction ($P = 0.938$) observed for microbial crude protein levels, and HRUP treatment did not affect its values ($P = 0.189$) (Fig. 3B). Intriguingly, an increasing trend toward week has been noted for MCP levels irrespective of treatment ($P = 0.028$).

**Alpha diversity indices increased with time, and richness increased with HRUP treatment.** Initial analyses of species richness (observed species and Chao1) and sample diversity (Shannon and Simpson) for fecal microbiota showed that no significant treatment-week interaction ($P > 0.05$) was observed for these four alpha indices (Fig. 4A). Of note, week had a remarkable effect on these indices, as indicated by a considerable gradual increment from week 0 to week 8 after weaning in both treatments ($P < 0.01$). Moreover, compared to the LRUP group, the HRUP group had greater species richness ($P < 0.01$) but similar species diversity ($P > 0.05$). The inferred microbial function and metabolite analysis based on SCFA profile both confirmed that HRUP treatment enhanced butyrate production. The enrichment of only *Eubacterium* by HRUP treatment highlighted its potential in butyrate production.

**Week and treatment drove the segregation of fecal microbiota.** The 16S rRNA data from fecal samples were used to generate principal coordinate axis plots (PCoA) for two treatments at each week using Bray–Curtis beta-diversity measurements. An Adonis test revealed no significant interaction between treatment and week ($P = 0.932$) in fecal community structure. Intriguingly, both week ($P < 0.001$) and treatment ($P = 0.001$) altered the microbiome configurations extensively (Fig. 4B). Pairwise permutational multivariate analysis of variance (PERMANOVA) yielded significant differences ($P < 0.01$) in bacterial diversity among 9 weeks of irrespective of treatment. Moreover, a temporal separation of bacterial configurations was detected between HRUP and LRUP treatments; this separation was statistically significant until week 6 after weaning and remained statistically significant during subsequent weeks 7 and 8 ($P < 0.05$, PERMANOVA) (Fig. 4C).

**Discovery of gut microbial biomarkers associated with dietary treatments.** As mentioned above, the separation of fecal bacterial configurations between two treatments was significant only at weeks 6, 7, and 8 ($P < 0.05$), and there was no significant effect of week on fecal bacterial community at these three time points. Therefore, the

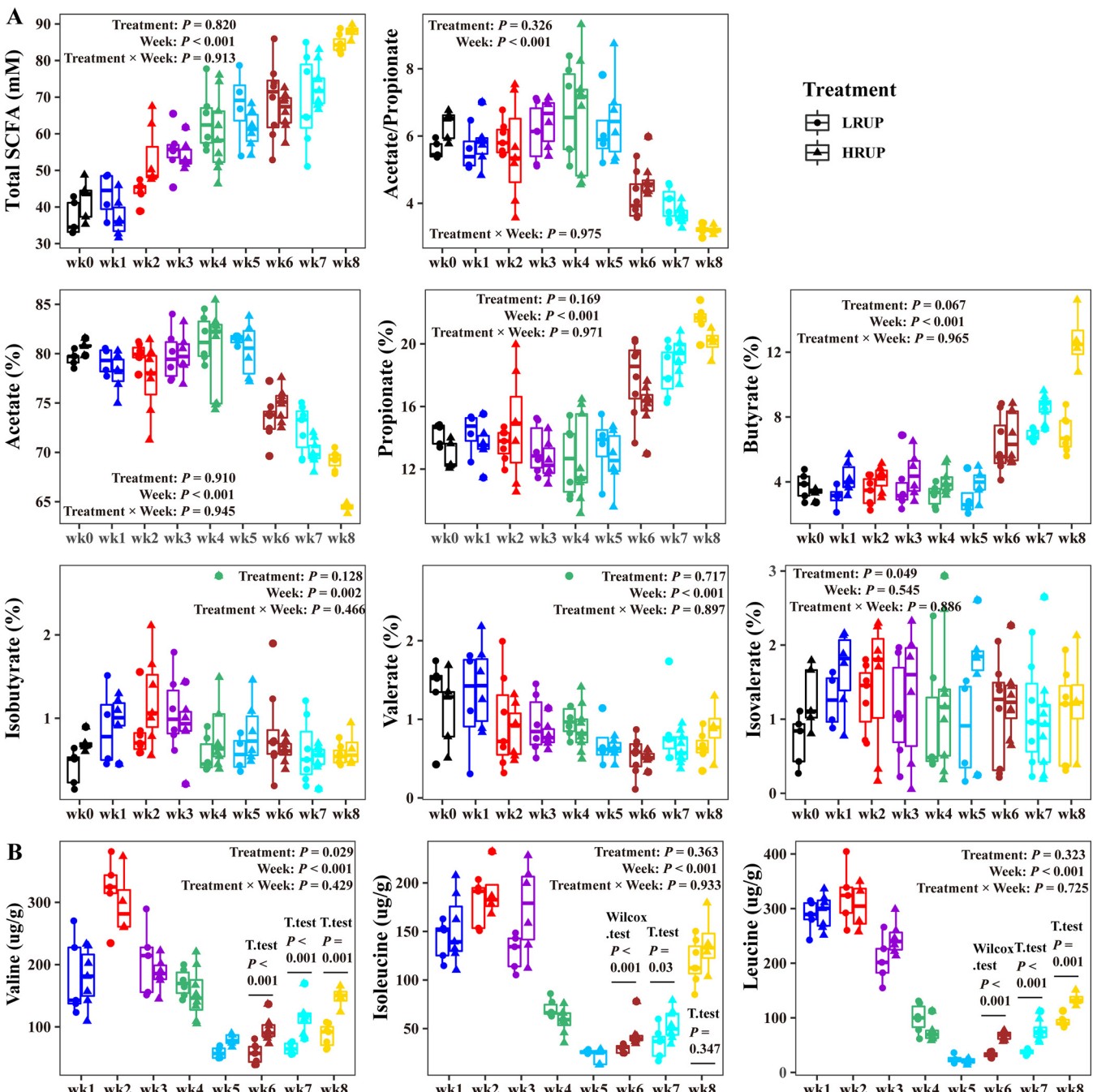

**FIG 2** Microbial metabolic products in the feces of goats fed with HRUP or LRUP diets at nine time points after weaning. (A) Total short-chain fatty acid (SCFA) concentration and molar proportion of acetate, propionate, butyrate, isobutyrate, valerate, and isovalerate. (B) Concentrations of branched-chain amino acids (valine, isoleucine, leucine).

effect of treatment on bacterial composition and predicted 3rd level KEGG pathways was analyzed using a two-group White's nonparametric *t* test with weeks 6, 7, and 8 as repeats in STAMP. The extended error bar plot for the difference between HRUP and LRUP treatments regarding distinctive diet-driven microbial communities included three genera, with *Eubacterium* significantly enriched ($P = 0.023$) in HRUP and *Faecalibacterium* and *Bifidobacterium* markedly boosted ($P < 0.05$) in LRUP (Fig. 5A). In terms of predicted microbial function, five 3rd-level KEGG pathways, "Bacterial motility proteins," "ABC transporters," "Flagellar assembly," "Bacterial chemotaxis," and "Butanoate metabolism," were enriched ($P < 0.05$) in the feces of HRUP goats, while three pathways, including "Citrate

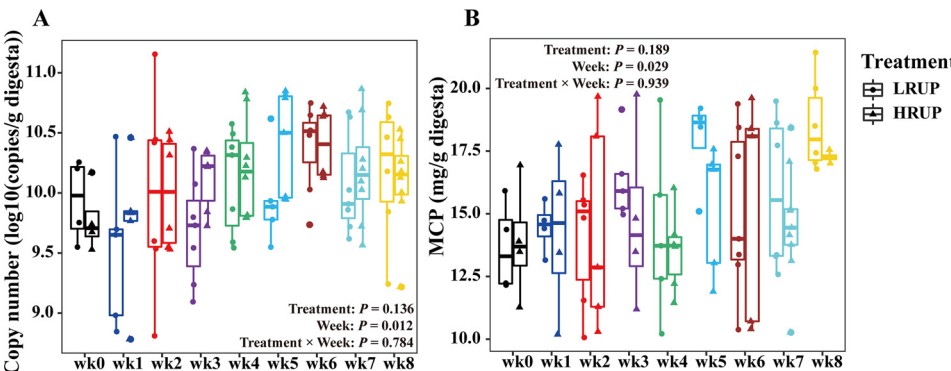

**FIG 3** Copy numbers (log$_{10}$ [copies/g digesta]) of 16S rRNA genes of total bacteria (A) and the concentration of MCP (mg/g digesta) (B) in the feces of goats fed with HRUP or LRUP diets at nine time points after weaning.

cycle (TCA cycle)," "Glycolysis/Gluconeogenesis," and "Lysosome," were more abundant ($P < 0.05$) in the feces of LRUP goats (Fig. 5B).

**Microbiome configurations and predicted functions revealed a temporal pattern of succession irrespective of dietary treatments.** We longitudinally tracked the reassembling process of fecal microbiota after weaning in goats. Intriguingly, both the between-group (Fig. 4B) and within-group (Fig. 6A) Bray–Curtis distances revealed three distinct phases of progression after weaning in the characteristics of fecal microbiota (i.e., early phase [weeks 0 to 2], middle phase [weeks 3 to 6], late phase [weeks 6 to 8]). On one hand, the pairwise PERMANOVA indicated that bacterial diversity in the early, middle, and late phases were different from each other ($P < 0.01$) (Fig. 4B). On the other hand, a wave crest appeared in within-group dissimilarity during middle phase, where the Bray–Curtis dissimilarity among different individuals within each

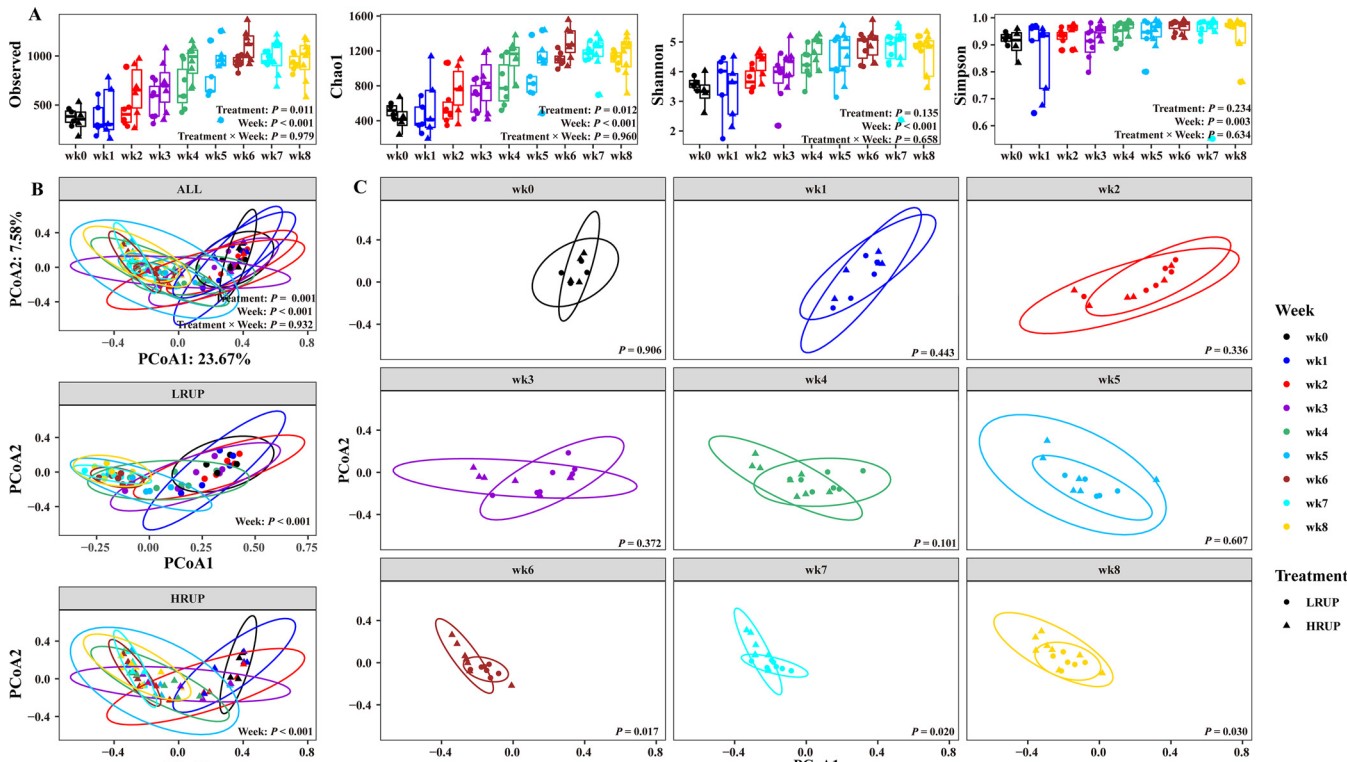

**FIG 4** Alpha diversity and beta diversity in the fecal microbiome of goats fed with HRUP or LRUP diets. (A) Species observed, Chao 1, Shannon, and Simpson. (B) Principal coordinate analysis (PCoA) of bacterial community structure at nine time points after weaning, including groups of LRUP and HRUP, only LRUP, and only HRUP. (C) PCoA of bacterial community structure between two treatments (HRUP versus LRUP) at each week.

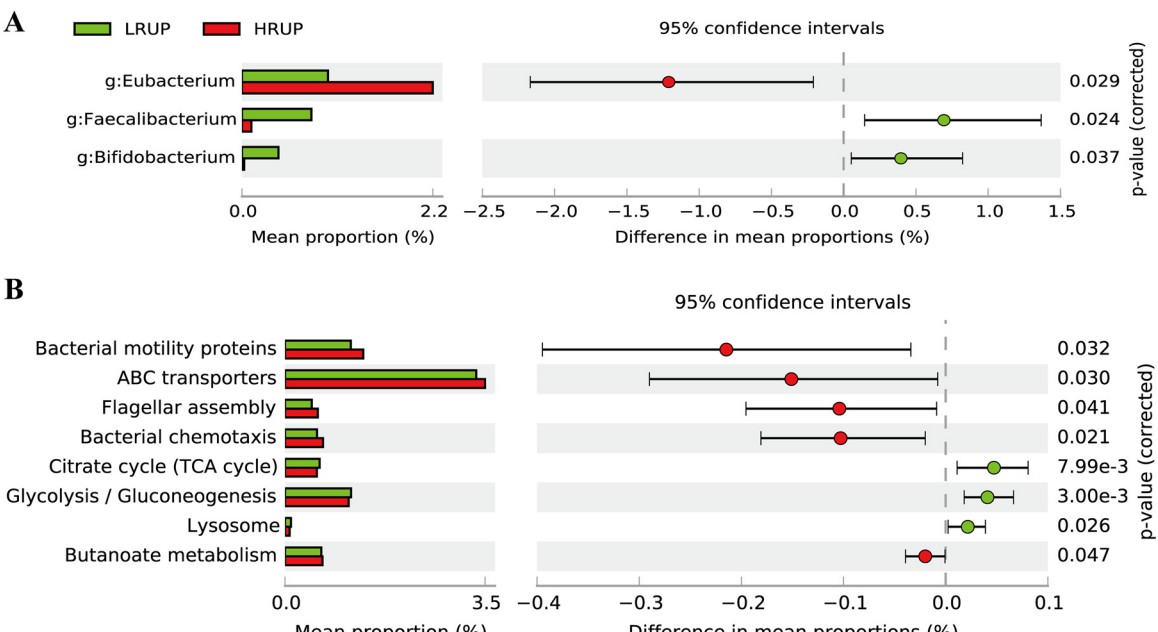

**FIG 5** Extended error bar plot identifying significant differences between mean proportions of bacterial taxa at the genus level (A) and the predicted functions of the 3rd level KEGG pathways (B) in LRUP (green) and HRUP (red) goats. Corrected *P* values are shown at right.

week was the highest (Fig. 6A). This indicted that in response to solid feed inclusion, the fecal microbiota exhibited a convergent-discrete-convergent trajectory. Further taxonomic analysis indicated that the fecal microbiome exhibited a temporal structure after weaning (Fig. 6B). Initial fecal microbiota at the genus level was typically dominated by *Bacteroides* (3.8% to 49%), *Escherichia/Shigella* (1.1% to 19%), *Bifidobacterium* (0.6% to 49%), *Sharpea* (0.3% to 5.1%), and *Faecalibacterium* (0.2% to 9.7%), but their abundance decreased with week (Fig. 6C). *Ruminococcus*, *Methanobrevibacter*, and *Eubacterium* were the minor bacterial linkages (<5%) until week 4 after weaning, but their abundance increased thereafter.

Among the 123 predicted KEGG pathways (average abundance > 0.01%), relative abundances of 19 pathways and 33 pathways changed with age for HRUP and LRUP goats, respectively (Table S1). Notably, 5 carbohydrate-related pathways, including amino sugar and nucleotide sugar metabolism, ascorbate and aldarate metabolism, galactose metabolism, glyoxylate and dicarboxylate metabolism, and pentose and glucuronate interconversions, were reduced gradually in both groups as goat aged after weaning. Butanoate metabolism pathway was in higher abundance in elder HRUP goats, while it was unchanged with week in LRUP goats. One lipid-related pathway, lipid biosynthesis proteins, was remarkably enhanced as goats aged in both groups. Three pathways related to amino acid metabolism changed significantly, with cysteine and methionine metabolism and phenylalanine, tyrosine, and tryptophan biosynthesis increased, while tryptophan metabolism decreased with age in LRUP goats. Except for nutrient metabolism, 3 pathways conducive to cell motility, including bacterial chemotaxis, bacterial motility proteins, and flagellar assembly, were more abundant with week. Moreover, the pathways of phosphotransferase system (PTS) and bacterial toxins were in lower abundance in elder goats in both HRUP and LRUP groups.

## DISCUSSION

Engineered systems for rearing humans and those for rearing animals both include phased feeding programs that smoothly transition them from early breast milk diets with high nutritional values to later solid diets with less-digestible components (13, 21). Achieving predictable robust weight gain and desirable health status at an affordable price

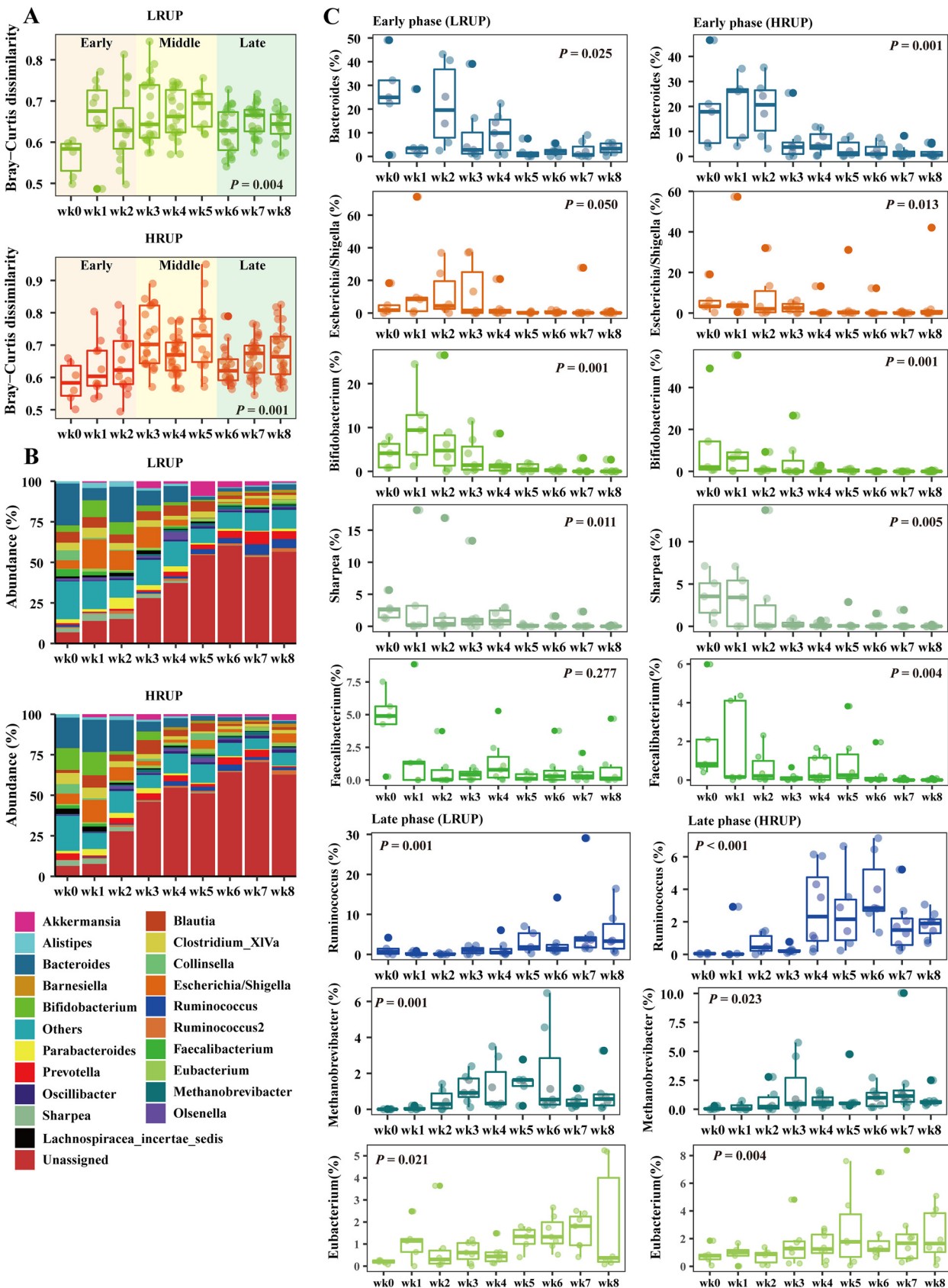

**FIG 6** Overview of temporal progression of fecal microbiota at nine time points irrespective of dietary treatments. (A) Bray–Curtis derived within-group dissimilarity matrices that revealed three distinct phases of progression, early phase (weeks 0 to 2), middle phase (weeks 3 to 5),

requires diets formulated based on a systematic understanding of when, which, and how the gut microbiota plays a key role in nutrient processing and energy harvest (5, 21). Our focus on the MP profile model in weaned goats is linked to deliberate precise manipulation of key growth- and health-promoting gut microbial features of young mammals (pigs and children). The first novel finding of this study is that at least 6 weeks of dietary manipulation are required to distinguish the beneficial effects of HRUP from LRUP treatment in weaned goats, emphasizing that the duration must be deliberated in order to reach its optimal effect in ruminant livestock. Similarly, the structural and functional adaptation of the pigs' fecal microbiome in response to dietary changes also lasts for 3 to 4 weeks (22) despite the usually assumed time span of several days. We further provide evidence that HRUP intervention considerably improved the metabolic efficiency (growth performance and nutrient digestibility) of weaned goats, consistent with previous observations in steers (23) and lambs (24). It has been proposed that HRUP intervention would elicit increment in available amino acids' flow to the small intestine and improved amino acid balance and, therefore, presumably promoted nutrient digestion (24, 25). Given that serum amino acid (AA) levels represent the net result between MP supply and AA utilization (19), the elevated levels of several serum AAs, BCAAs in particular, strongly confirm an improved AA balance by HRUP intervention in weaned goats.

A growing body of evidence suggests that gut microbiome-expressed functions have profound implications for animal growth, health, and productivity (26). Hence, an integrated analysis of microbial composition, metabolites, and predicted functions was conducted to elucidate the underlying mechanisms during HRUP intervention. As expected, a shift in favor of *Eubacterium*, which possess the capacity to produce butyrate through the acetyl coenzyme A (acetyl-CoA) pathway, was identified in response to HRUP intervention (27). Further investigation of fecal microbial metabolites uncovered a boost in gut butyrate production and a decline in acetate production in HRUP goats, which was in accordance with the enhanced butanoate metabolism pathway in microbial functions (28). Synchronously, a prominent increase in fecal BCAA levels was noted for HRUP group, which might contribute to an important source for the above-mentioned elevated serum BCAAs. Since gut microbiota can both produce (e.g., *Prevotella copri* and *Bacteroides vulgatus*) and use (e.g., *Butyrivibrio crossotus* and *Eubacterium siraeum*) BCAAs (21), the greater fecal BCAA levels represented a net result from the bacteria biosynthesis and catabolism of BCAAs. It is well established that BCAAs share common biosynthetic pathways all stemming from intermediates from pyruvate metabolism (29), a central component of the tricarboxylic acid (TCA) cycle, which perfectly links the metabolism of carbohydrates, amino acids, and lipids. Taken together, HRUP intervention elicited a shift from acetate production toward butyrate and BCAA production involving the enrichment of gut *Eubacterium* members.

Survival of gut bacteria depends on their capacity to monitor and respond to the ever-changing environmental conditions, from the levels of nutrients, toxins, and oxygen to pH, temperature, and osmolarity (30). A prominent way to respond is bacterial chemotaxis, featured by precisely migrating toward the beneficial chemical stimuli and away from a toxic chemical substance (30, 31). As anticipated, HRUP intervention promoted the bacterial chemotaxis pathway, which might elicit a more efficient move toward the food source, such as amino acid and sugar in fecal microbiota. Increasing evidence indicates that the flagellum is the motor organelle for bacterial chemotaxis, and its assembly is in a stepwise manner, with the basal body assembled first, followed by the hook and finally the filament (32, 33). Hence, the enriched flagellar assembly in response to HRUP intervention provided a premise for the above-mentioned effective motility toward food source. After reaching the food, the next step for gut microbiota is to pump dietary compounds across biological members, with the aid of transporters

**FIG 6** Legend (Continued)

and late phase (weeks 6 to 8). (B) Relative abundance of fecal microorganisms at the genus level in LRUP or HRUP goats. (C) Early phase fecal microbiota at the genus level was typically dominated by *Bacteroides*, *Escherichia/Shigella*, *Bifidobacterium*, *Sharpea*, and *Faecalibacterium*, while late phase fecal microbiota was featured by *Ruminococcus*, *Methanobrevibacter*, and *Eubacterium*.

from the ATP-binding cassette (ABC) superfamily (34). When functioning as importers, ABC transporters orchestrate a series of conformational changes, therefore resulting in translocation of various substrates, ranging from oligopeptides and oligosaccharides to small ions, across the membrane into cells (35). Therefore, it is not surprising to observe an elevation in the ABC transporters pathway in HRUP goats. To sum up, in response to HRUP intervention, specific gut *Eubacterium* members actively switched toward the nutrients using flagella and then scavenged nutrients using a suite of ABC transporters and inevitably triggered a shift toward butyrate and BCAA production.

The milk-oriented gut microbiome harvests dietary milk glycans, while the weaned gut microbiota harvests plant glycans in a pig model (36). Our dense longitudinally collected fecal samples expand this knowledge to further dynamic assemble trajectories in gut microbiota using weaning goats as a model. Similar to reports in children (37), the surge in solid feed consumption (16-fold increase), independent of MP profile, provided the fecal microbiota with more substrate availability, sequentially boosting their alpha and beta diversities. Furthermore, it is worth noting that the gut microbiome underwent the developmental trajectories of evolving from an immature environment that is dominated by *Bacteroides*, *Escherichia/Shigella*, *Bifidobacterium*, and *Sharpea* (early phase), through a dynamic and unstable stage (middle phase), and then to a mature environment with *Ruminococcus* and *Methanobrevibacter* enriched (late phase). In line with previous observation in the fecal microbiota of healthy infants (38), the high prevalence of *Escherichia/Shigella* in early phase contributed profoundly to provide anaerobic environment for other bacteria due to its scavenging oxygen ability. As a probiotic, the *Bifidobacterium* possessed the ability to utilize the fucose in milk as the major substrate and then convert pyruvate into formate, lactate, and acetate (27, 39). The high abundance of starch-degrading *Bacteroides* and lactic acid-producing *Sharpea* (40) observed in early phase jointly contribute the accumulation of SCFAs, especially acetate. In contrast, the gut environment in late phase was more favorable for the fiber-degrading anaerobes of *Ruminococcus* and *Eubacterium* (41), similar to previous reports in children (37) and young goats (7), and is likely conducive to the almost doubled accumulation of SCFAs as major microbial metabolites. Insights into BCAAs echoed the fluctuations in microbial metabolites with age, shedding light on the observation that the metabolic potentials of AAs also progressed sequentially as weaned goats aged (29). Intriguingly, the middle phase exhibited the highest individual variation, implying that the gut microbiome responded to the weaning diet in various ways, while stabilizing after 6 weeks. Similarly, young children (37) and calves (42) both have individual dynamics in the gut microbiota development trajectory. These highlight the priority of taking the high individual variation and dynamics of intestinal microbiome into account and performing personalized nutrition in young mammals.

In agreement with previous studies in dairy calves (15), our results also observed a remarkable reduction in several inferred KEGG pathways associated with carbohydrate metabolism in the feces microbiome as weaned goats aged, conducive to the shift in nutrient metabolism. Given that the gut microbiota relies on the phosphotransferase system (PTS) to efficiently import cognate carbohydrates into cells (43), a drastic decrease in the PTS pathway was thus noted as goats aged. Considering that several pathways related to lipid and AA metabolism also changed with age, we can propose that gut bacteria can integrate their nutritional status with diverse environmental stimuli (15, 43). Concurrently, it is further recapitulated that three pathways conducive to cell motility showed sequential progression in response to weaning. As mentioned above, most gut microorganisms rely on their elaborate systems to sense stimuli and alter the functioning of their motility machinery to migrate to their optimal nutritional gradient (33). Taken together, as weaned goats aged, fecal microbiome sequentially shifted their functions in a programmable manner with improvements in cell motility but reductions in carbohydrate metabolism. It is of particular importance to further validate and increase the power of our data through omics-based, culture-based, or transplantation-based analysis of intestinal microbiota.

In conclusion, a dense longitudinal characterization of compositions, metabolites, and inferred functions of fecal microbiota highlighted that at least 6 weeks of intervention of HRUP are needed to achieve far-reaching implications on the productive performance. Further insight into the possible underlying mechanism indicated that HRUP intervention promoted AA balance in the hindgut via favoring the proliferation of butyrate-producing *Eubacterium* members, facilitating the cell motility of gut microbiota toward nutrients such as glucose and AAs, incorporating into cells with the aid of ABC transporters, and shifting microbial metabolism from acetate toward butyrate and BCAA production to support body growth (Fig. 7A and B). Furthermore, similar gut microbiota mature trajectories were noted independent of MP profile, such as the enriching abundance of fiber-degrading *Ruminococcus*, the enhancement of microbial cell motility, and the shift of fermentation type, reflecting an inner law in the progression of gut microorganisms in weaned goats over time (Fig. 7C) and highlighting the importance of taking microbiota dynamics into account.

## MATERIALS AND METHODS

**Ethics approval.** All experimental procedures were approved by the Animal Care Committee, Institute of Subtropical Agriculture, The Chinese Academy of Sciences, Changsha, Hunan province, China (permission CAS2013020).

**Goats, diets, and experimental design.** Twenty-four kid goats (3 weeks old, 3.8 ± 0.2 kg) were selected as experimental animals and randomly stratified to one of the two solid feed dietary treatments in the form of powder with different metabolizable protein profiles (RUP and RDP ratios) (Table 2), HRUP diet (RUP: RDP = 50:50), and LRUP diet (RUP: RDP = 35:65). Dietary treatments were formulated to meet the recommendation of Lu et al. (44) for weaning goats in China. The RUP and RDP of diets were calculated based on their values of each feed ingredient in the China feed data (http://chinafeeddata.org.cn). Goats kids lived with their mothers for 1 week of adaptation to solid feed and would imitate their mothers' behavior to eat solid feed, although they ate very little. Subsequently, goats (4 weeks old) were weaned and housed in individual pens (width by length by height, 1.00 m by 1.20 m by 0.75 m) throughout the experiment and were fed either an HRUP or an LRUP diet. Goats were fed in total mixed diets twice daily at 8:00 and 17:00 h in amounts to ensure less than 10% orts (the amount of remaining feeds/the amount of feeding feeds). All weaned goats had free access to water throughout the experimental period. Feed refusal was recorded daily. The growth performance parameters (body weight, body height, body length, and chest grith) were measured every week from weaning until the goats were 12 weeks old (Fig. 1A).

**Sample collection.** At 0, 1, 2, 3, 4, 5, 6, 7, and 8 weeks postweaning (9 days in total), feces at the rectum of both groups of goats were sampled 4 h after the morning feeding (Fig. 1A). Two grams of feces was immediately frozen in liquid nitrogen and stored at −80°C in the lab until microbial DNA extraction. Approximately 0.2 g of feces was homogenized with 2 mL phosphate-buffered saline (pH = 7.4) and 0.2 mL of 25% (wt/vol) metaphosphoric acid, incubated at 4°C for 2 h, and centrifuged at 12,000 × $g$ for 10 min at 4°C. The supernatant was gathered and filtered through a 0.45-$\mu$m membrane for subsequent short-chain fatty acid analysis. About 0.2 g of feces was homogenized with 2 mL phosphate-buffered saline (pH = 7.4), incubated at 4°C for 2 h, and centrifuged at 12,000 × $g$ for 10 min at 4°C. The supernatant was collected for subsequent free amino acid analysis. Meanwhile, during weeks 6 and 8 after weaning, goat feces were gathered each day (14 days in total) and stored at −20°C for nutrient digestibility analysis (Fig. 1A). At the end of the experimental period, blood samples were collected from the jugular vein of each kid and centrifuged at 3,500 × $g$ for 15 min to separate serum. The serum samples were stored in plastic tubes frozen at −20°C until analysis of amino acid profile.

**Nutrient digestibility.** Acid-insoluble ash was used as an internal marker to measure nutrient-apparent total-tract digestibility and determined in a muffle furnace at 550°C for 8 h (45). Dry matter (DM), crude protein (CP), crude fiber (CF), neutral detergent fiber (NDF), and acid detergent fiber (ADF) were measuring using the methods of AOAC (46). Briefly, feces were dried and ground to pass through a 1-mm sieve and analyzed for DM (105°C 24 h). The CP content in samples was calculated from total nitrogen content by multiplying a conversion factor, usually 6.25. Analysis of CF, NDF, and ADF was carried out using the filter bag technique, and ether extract (EE) content of the samples was determined by SOX416 (Gerhardt, Germany) with the Twisselman method, using diethyl ether as a solvent.

**Free amino acid profiles in serum and feces.** Serum and fecal samples were mixed thoroughly with 1 mL of 8% sulfosalicylic acid solution and incubated at 4°C for 12 h (47). After centrifugation at 13,000 rpm for 15 min, the supernatant was filtered through 0.22-$\mu$m filters and then analyzed by an ion-exchange amino acid analyzer (L8800, Hitachi, Tokyo, Japan).

**Determination of SCFA and MCP concentration.** The SCFA concentrations (acetate, propionate, butyrate, isobutyrate, valerate, and isovalerate) were determined using a gas chromatographer (Agilent 7890A, Agilent Inc., Palo Alto, CA), according to the method described by Jiao et al. (13). The content of microbial crude protein (MCP) in fecal samples was determined using the method outlined in our previous study (47).

**DNA extraction and total bacteria quantification.** Fecal samples (approximately 0.2 g) were homogenized using a bead beater, and then the total DNA was extracted using the QIAamp DNA stool

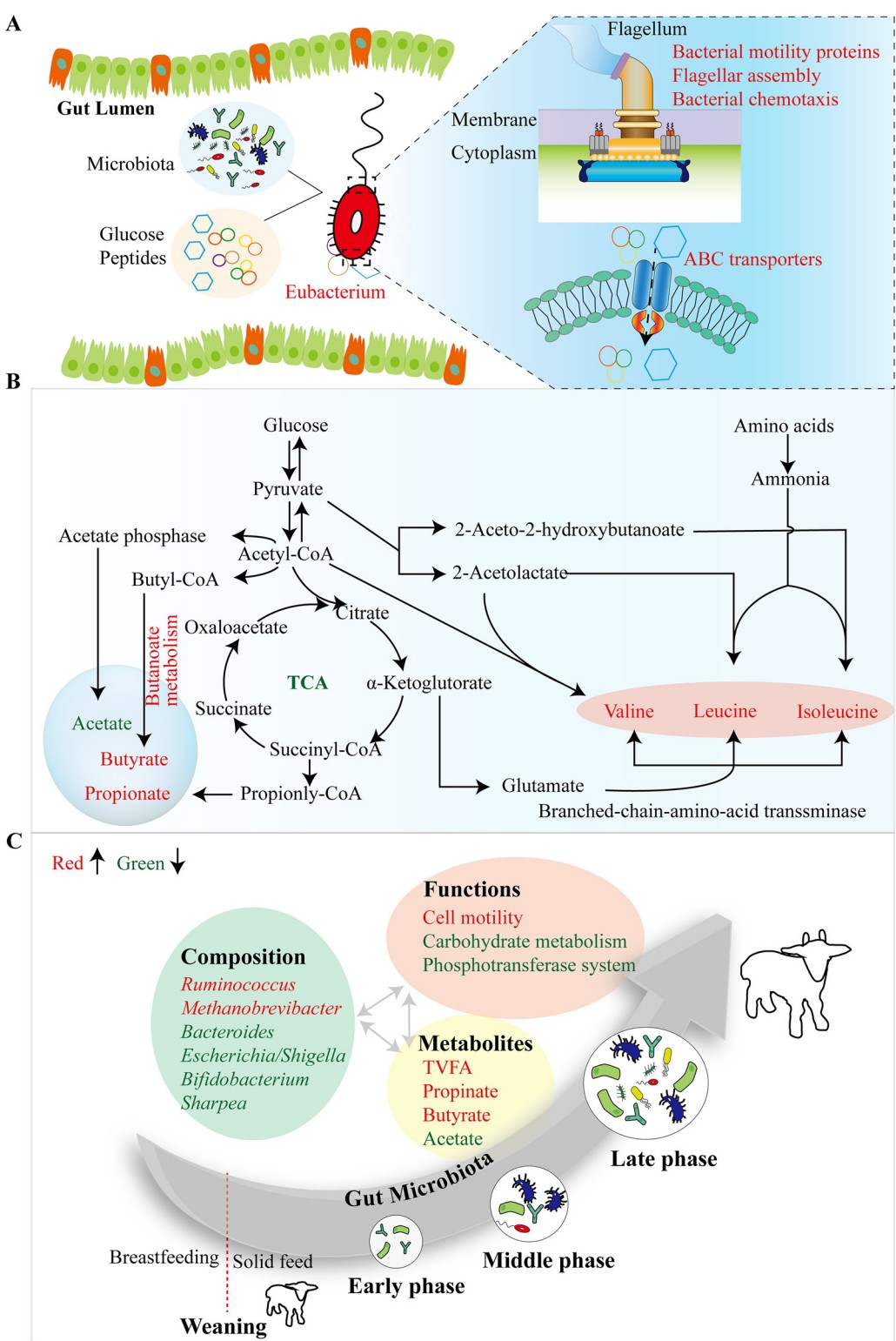

**FIG 7** Integrative diagram showing the main results from the present work. (A and B) Proposed mechanism of actions driven by fecal microbiota during HRUP intervention. HRUP intervention induced an enrichment in *Eubacterium*, promoted its switch toward nutrients using flagella through cell motility, and then scavenged nutrients using a suite of ABC transporters (A). In bacterial cells, HRUP enhanced butanoate metabolism pathway and shifted microbial metabolism from acetate production toward butyrate and BCAA (valine, leucine, and isoleucine) production (B). (C) Temporal succession pattern of fecal microbiota (composition, functions, and metabolites) irrespective of dietary treatments. Red font means an increase, while green font represents a decrease.

**TABLE 2** Feed ingredients and chemical composition of experimental diets

| Diet composition item | Value for treatment[a]: | |
|---|---|---|
| | LRUP | HRUP |
| Ingredient, % | | |
| Alfalfa | 30.00 | 30.00 |
| Corn grain | 32.70 | 35.00 |
| Fat powder | 1.62 | 4.37 |
| Wheat middling and red dog | 8.00 | 0.00 |
| Corn gluten meal | 0.00 | 18.00 |
| Soybean meal | 15.30 | 0.00 |
| Calcium carbonate | 0.18 | 0.13 |
| Dicalcium phosphate | 2.70 | 3.00 |
| Premix[b] | 2.00 | 2.00 |
| Salt | 0.50 | 0.50 |
| Lactose | 7.00 | 7.00 |
| Chemical composition[c], % | | |
| Metabolizable energy, Mcal/kg | 2.89 | 2.80 |
| Neutral detergent fiber | 19.51 | 20.40 |
| Crude protein | 16.55 | 16.47 |
| Rumen-degradable protein | 10.63 | 8.34 |
| Rumen-undegradable protein | 5.84 | 8.21 |
| RUP:RDP ratio, (% of dietary CP) | 35:65 | 50:50 |
| Calcium | 1.36 | 1.38 |
| Phosphorous | 0.94 | 0.94 |

[a]Treatments were (i) low RUP/RDP ratio (LRUP) = 35:65 and (ii) high RUP/RDP ratio (HRUP) = 50:50.
[b]Contained per kilogram of supplement: 5.000 g $FeSO_4 \cdot H_2O$, 1.984 g $CuSO_4 \cdot 5H_2O$, 0.550 g $CoCl_2$, 0.750 g $KIO_3$, 10.169 g $MnSO_4 \cdot H_2O$, 7.042 g $ZnSO_4 \cdot H_2O$, 0.250 g $NaSeO_3$, 0.033 g vitamin A and 0.006 g vitamin D.
[c]Calculated based on www.chinafeeddata.org.cn.

mini kits (Qiagen GmbH, Hilden, Germany) according to protocols detailed in our previous study (7). The quality of DNA extracts was assessed using 1% agarose gel electrophoresis, and the quantity of DNA was tested on the basis of absorbance at 260 and 280 nm using a Nano Drop 2000 spectrophotometer (Thermo Scientific, Waltham, USA).

The quantitative real-time PCR analysis of total bacteria was carried out with an ABI 7900 sequence detection system (Applied Biological System, Foster, CA, USA), using SYBR green premix Pro *Taq* HS qPCR kit AG11701 (Accurate Biotechnology [Hunan] Co., Ltd.). The specific test procedures were in accordance with Jiao et al. (13).

**High-throughput sequencing and bioinformatics analysis.** The V3 and V4 regions of the bacterial 16S rRNA gene were used for PCR amplification with the primers 341F (5'-CCTAYGGGRBGCASCAG-3') and 806R (5'-GGACTACNNGGGTATCTAAT-3'). The procedures and conditions of PCRs were performed according to the methods described by Jiao et al. (7). The PCR products were purified using the AxyPrep DNA gel extraction kit (Axygen Biosciences, Union City, CA, USA) according to the manufacturer's instructions. Amplicons were quantified using Qubit3.0 fluorometrically, normalized, and pooled at equimolar ratios prior to sequencing with the Illumina HiSeq platform.

The barcodes and sequencing primers in the raw data were removed before data processing. The sequences with a similarity level of more than 99% were clustered into amplicon sequence variants (ASVs) using the method of unoise3 by USEARCH v.11.0.667 (48) based on the reference database of SILVA v.132 (49). Afterwards, the representative sequences were submitted to the RDP classifier v.16 (50) to obtain the taxonomy assignment with a 0.80 confidence threshold. Also, before analyzing, singletons (ASVs with only one sequence) were removed. Analyses of alpha and beta diversities were performed in R using packages vegan v 2.5.7 (51) and Phyloseq v 1.34.0 (52).

**Functional prediction of microbial pathway abundances by PICRUSt.** The PICRUSt (Phylogenetic Investigation of Communities by Reconstruction of Unobserved States) was used to infer functional potentials of fecal microorganisms using the ASVs table (53). First, normalize_by_copy_number.py script was carried out to normalize the ASVs. Second, the normalized ASVs table was input into the predict_metagnomes.py script to generate predicted microbial pathway abundances based on Kyoto Encyclopedia of Genes and Genomes (KEGG).

**Statistical analysis.** Values of body weight, body parameters, apparent digestibility, and serum amino acid profile between two treatments were checked for normality and analyzed using two-tailed Student's *t* tests allowing for equal or unequal variance in R software (version 4.0.4) (54). The differences in the total bacteria copy number and MCP were assessed by two-way analysis of variance (ANOVA) of treatment, week, and their interaction and followed by a *post hoc* Tukey's honestly significant difference (Tukey's HSD) test. As values of fecal SCFA profiles, AA profiles, and alpha diversity measures did not conform to the normal distribution, they were analyzed using Scheirer-Ray-Hare test (a nonparametric test used for a two-way factorial design) in R "rcompanion" package of treatment, week, and their

interaction, followed by a *post hoc* Dunn's test in R "PMCMRplus" package with false-discovery rate (FDR) correction for multiple comparisons (55, 56).

Beta-diversity measures were visualized using principal coordinate analysis (PCoA) and tested using permutational analysis of variance (PERMANOVA) implemented using the Adonis function in R "vegan" package with the Bray–Curtis method to calculate pairwise distances and 999 permutations (51). The statistical differences of the dominant genera (top 20) and predicted KEGG level 3 pathways (average abundance > 0.01%) were conducted using STAMP (Statistical Analysis of Metagenomic Profiles v2.1.3) (57). Differential genera and pathways among weeks within each treatment were analyzed using Kruskal-Wallis H-test, followed by a *post hoc* Tukey-Kramer's test with FDR correction for multiple comparisons. The effect of treatment on KEGG level 3 was analyzed using a two-group White's nonparametric *t* test with weeks 6, 7, and 8 as repeats and visualized by extended error bar in STAMP.

**Data availability.** The data that support the findings of this study are openly available at NCBI SRA PRJNA780611.

## SUPPLEMENTAL MATERIAL

Supplemental material is available online only.

**SUPPLEMENTAL FILE 1,** XLSX file, 0.5 MB.

## ACKNOWLEDGMENTS

This study was supported by the National Natural Science Foundation of China (no. 31730092, U20A2057) and Strategic Priority Research Program of the Chinese Academy of Sciences (grant no. XDA26040304).

We confirm that we have no conflicts of interest.

J.J. and T.Z. designed the research, W.J. and Z.X. performed the research and analyzed the samples. W.J., W.M., Z.C., and J.J. contributed intellectually to the analysis and interpretation of the data. W.J., J.J., and T.Z. wrote the manuscript. All authors read and approved the final version of the manuscript.

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
