## [Reviewer comments · Microbiology Spectrum]

Microbiology Spectrum

Enhancing metabolic efficiency through optimizing metabolizable protein profile in a time progressive manner with weaned goats as a model: involvement of gut microbiota

Jian Wu, Xiaoli Zhang, Min Wang, Chuanshe Zhou, Jinzhen Jiao, and Zhiliang Tan

Corresponding Author(s): Jinzhen Jiao, Institute of Subtropical Agriculture, CAS

Review Timeline:

Submission Date:	December 7, 2021
Editorial Decision:	January 16, 2022
Revision Received:	March 3, 2022
Accepted:	March 16, 2022

Editor: Jinxin Liu

Reviewer(s): Disclosure of reviewer identity is with reference to reviewer comments included in decision letter(s). The following individuals involved in review of your submission have agreed to reveal their identity: Junhua Liu (Reviewer #1); Zhigang Zhang (Reviewer #2)

Transaction Report:

DOI: <https://doi.org/10.1128/spectrum.02545-21>

January 16, 2022

Dr. Jinzhen Jiao
Institute of Subtropical Agriculture, CAS
Changsha, Hunan 410125
China

Re: Spectrum02545-21 (Enhancing metabolic efficiency through optimizing metabolizable protein profile in a time progressive manner with weaned goats as a model: involvement of gut microbiota)

Dear Dr. Jinzhen Jiao:

Link Not Available

Sincerely,

Jinxin Liu

Journals Department
Reviewer comments:

Reviewer #1 (Comments for the Author):

Precise dietary intervention of nitrogen metabolism in early life has significant implications in the long-life productivity and health of young ruminants and lowering their environmental footprint. In the present study, Wu et al. investigated when, which and how the faecal microbiome responded to metabolizable protein profile intervention in young ruminants. The topic is of interest, the manuscript is well written and those findings have some significance for ruminant production, but some minor revisions should be done.

Line 380: How did you determine RDP and UDP ?

Line 385: All the kid goats were weaned at 4 weeks old. Is it a normal strategy in goat production?

Line 392: The digestibility measures are not described in enough detail. How many days were feces collected? How were they kept separate from urine?

How big are the pens? Were they maintained in the pens during the digestibility measures?
When were feed samples collected for what you analyzed in Table 1?

Line 392: What time were the samples collected after feeding? The sampling time would affect microbial composition and metabolites.

Table 1: The processing pattern (meal or pellet?) of diet should be provided.

Reviewer #2 (Comments for the Author):

The authors provide a valuable study for precise feeding of goat by performing a series of time-scale experiments. Some findings from growth performance and metabolic phenotypes were very interesting. For example, Why HRUP can increase amino acid contents (i.e., Lys, His, Val, Phe in Table 2) in serum? It is surprising. High rumen undegradable protein diet (HRUP) remarkably improved the performance of goats (Fig.1). Why? These phenomena are very interesting. However, there is only weak supports from microbial analysis. So, it is necessary to improve microbial analysis for obtaining enough evidence supporting observed phenotypes. Some comments as followed for the improvement of MS.

1. Experimental diets (Table 1): how to define RUP and RDP based on the chemical composition of experimental diets? The observed differences of many compositions were shown in Table 1. It is very difficult to understand the contribution of HRUP or LRUP or to control the effect of non-protein compositions to all results.
2. In Fig.6, the observed pattern may be associated with goat development but not HRUPs or LRUPs. See Figure 2 (SCFA), Figure 3 (Microbial counts and proteins), and Figure 4, significant differences were observed between weeks but not between treatments, although the difference of alpha-diversity indices (Richness & Chao1) between treatments. Such difference may be due to the production of many singleton ASVs but not from diet changes.
3. Figure 5: the results are very interesting. Thus, how to link the results of 5B to three bacterial genera (Eubacterium, Faecalibacterium, and Bifidobacterium) in 5A? why not showing the progression of those three bacteria genera over time? It is crucial to determine their roles. In contrast, some other bacteria except for Bifidobacterium were shown in Figure 6C. It is confusing. Whether some metagenomic evidence can be provided to confirm the contributions of Eubacterium, Faecalibacterium, and Bifidobacterium to some distinct metabolic phenotypes. Here, it is unclear why only Eubacterium was highlighted in the graph in Figure 7 but exclude Faecalibacterium, and Bifidobacterium.
4. Figure 6: it is arbitrary to divide time points into three phases because no obvious pattern was observed, for example (6A), for HRUP treatment, a clear decrease occurred in wk4 and reached to the levels of wk7 and wk8. Correspondingly, in Figure 6C, most of bacterial genera did not show similar patterns like those in Figure 6A. Hence, please refine those results.
5. Fig.7 showed charming models or pathways but related evidence needs to be further strengthened. Here, there is over-interpretation for microbial analysis.

Staff Comments:

Preparing Revision Guidelines

Please return the manuscript within 60 days; if you cannot complete the modification within this time period, please contact me. If you do not wish to modify the manuscript and prefer to submit it to another journal, please notify me of your decision immediately so that the manuscript may be formally withdrawn from consideration by Microbiology Spectrum.

Dear editors and reviewers,

Thanks for your letter and the reviewers' comments concerning our manuscript. Those comments are all valuable and very helpful for revising and improving the readability and quality of our manuscript. We have studied these comments carefully and made modifications which we hope to meet with your approval in a point-to-point manner. Revised parts were marked using a red font in the manuscript.

Responses to Reviewer #1

Precise dietary intervention of nitrogen metabolism in early life has significant implications in the long-life productivity and health of young ruminants and lowering their environmental footprint. In the present study, Wu et al. investigated when, which and how the faecal microbiome responded to metabolizable protein profile intervention in young ruminants. The topic is of interest, the manuscript is well written and those findings have some significance for ruminant production, but some minor revisions should be done.

Reply: Thanks a lot for your positive and constructive comments. This manuscript has been revised substantially according to the reviewer's suggestions. All the revisions have been highlighted using a red font in the revised manuscript.

Line 380: How did you determine RDP and UDP?

Reply: The RDP and RUP of these two diets were calculated based on the database of China feed ingredients (<http://chinafeeddata.org.cn>). Briefly, we queried the RDP and RUP content of each feed ingredient from the database, then calculated according to the ratio. Please see Lines 390-393 in the revised version.

Line 385: All the kid goats were weaned at 4 weeks old. Is it a normal strategy in goat production?

Reply: Thanks for your inquiry. In goat production, there are two ways of weaning strategies, i.e., early weaning and natural weaning. The natural weaning is prevalent for grazing ruminants, whereas early weaning (at 4 weeks) is usually carried out in intensive goat production to achieve economic and health benefits.

Line 392: The digestibility measures are not described in enough detail. How many days were feces collected? How were they kept separate from urine? How big are the pens? Were they maintained in the pens during the digestibility measures? When were feed samples collected for what you analyzed in Table 1?

Reply: Thanks for your detailed questions. Firstly, the digestibility measures have been described in detail. Please see Lines 419-421, 423-428 in the revised manuscript.

Secondly, faeces were collected at the beginning of the formal trial, and then once a week (9 days). Moreover, in order to accurately measure the digestibility, fecal samples were collected every day in the sixth (7 consecutive days) and eighth (7 consecutive days) weeks. Therefore, faeces were totally collected 23 days. Please see Lines 403-405, 413-415.

Thirdly, the feces and urine collection devices were composed of two parts, the upper net, which was used to collect faeces and the lower tray, which was used to collect urine. Throughout the experiment, kid goats were maintained in the pens and the size of the pen was 1.0 m × 1.2 m × 0.75 m (width × length × height). Please see Lines 395-397.

Finally, the feed formulas were designed at the beginning of the trial according to the theoretical guidance of scientific technology of feeding goat (China Agriculture Press, Beijing, China). And the chemical compositions were calculated based on <http://chinafeeddata.org.cn>. Please see Lines 390-393 and the note ³ to the Table 1.

Line 392: What time were the samples collected after feeding? The sampling time would affect microbial composition and metabolites.

Reply: Faeces at the rectum of both groups of goats were sampled about 4 h after the morning feeding. Please to see Lines 403-405 in the revised manuscript.

Table 1: The processing pattern (meal or pellet?) of diet should be provided.

Reply: The processing form of feed is powder. Please to see Line 388.

Responses to Reviewer #2

The authors provide a valuable study for precise feeding of goat by performing a series of time-scale experiments. Some findings from growth performance and metabolic phenotypes were very interesting. For example, Why HRUP can increase amino acid contents (i.e., Lys, His, Val, Phe in Table 2) in serum? It is surprising. High rumen undegradable protein diet (HRUP) remarkably improved the performance of goats (Fig.1). Why? These phenomena are very interesting. However, there is only weak supports from microbial analysis. So, it is necessary to improve microbial analysis for obtaining enough evidence supporting observed phenotypes. Some comments as followed for the improvement of MS.

Reply: Thank you for your interest in this manuscript, as well as insightful and constructive comments. Please let us clarify in detail. Firstly, we observed interesting phenotypes that HRUP improved growth performance and nutrient apparent digestibility. A notable increase in branch-chain amino acid contents further suggested HRUP improved serum amino acid balance. Considering the significance of gut microbiota in host performance and health, we conducted integrated analysis of bacterial diversity, metabolites and their inferred function. Our findings systematically reported when, which and how the faecal microbiome responded to metabolizable protein profile intervention in young ruminants.

However, your detailed comments really helped us a lot in improving the MS. We have seriously thought about them and provided our detailed responses below.

1. Experimental diets (Table 1): how to define RUP and RDP based on the chemical composition of experimental diets? The observed differences of many compositions were shown in Table 1. It is very difficult to understand the contribution of HRUP or LRUP or to control the effect of non-protein compositions to all results.

Reply: Thanks for your inquiry. The RDP and RUP of these two diets were calculated based on the database of China feed ingredients (<http://chinafeeddata.org.cn>). Briefly, we queried the RDP and RUP content of each

feed ingredient from the database, then calculated according to the ratio. Please see Lines 390-393 in the revised version.

Besides, previous studies indicated that optimizing metabolizable protein profile with RUP/RDP ratios could potentially improve nitrogen efficiency and decrease environmental pollution in adult ruminants. In this study, we aimed to conduct a dietary nitrogen intervention in young ruminants. These diets are formulated to adjust the RDP/RUP ratio as much as possible, while ensuring that the metabolizable energy, total crude protein content and other non-protein compositions are consistent between two treatments. Here we need to apologize for the mistake in Table 1, in which the NDF for LRUP is 19.51%. Hence, the non-protein compositions such as NDF are similar in both groups. Therefore, HRUP intervention contributed to most of the results. Of course, both diets meet the nutritional requirements of weaned goats.

2. In Fig.6, the observed pattern may be associated with goat development but not HRUPs or LRUPs. See Figure 2 (SCFA), Figure 3 (Microbial counts and proteins), and Figure 4, significant differences were observed between weeks but not between treatments, although the difference of alpha-diversity indices (Richness & Chao1) between treatments. Such difference may be due to the production of many singleton ASVs but not from diet changes.

Reply: Thanks for your inquiry and sorry to make you confused. Please let us clarify. Firstly, we have already removed singleton ASVs from the dataset since these singletons could be due to sequencing artefacts during microbial analysis, so such difference was not due to the production of many singleton ASVs. Please see Lines 466-467 in the revised manuscript.

Secondly, we cannot agree with you more that weeks might serve as the most significant driver for the SCFAs, microbial counts, MCPs and alpha-diversity indices. Despite these, it is notable that HRUP drastically elevated the molar percentage of butyrate in wk8 ($P < 0.05$, see Lines 141-144 in the revised manuscript), and significantly elevated ($P < 0.01$) faecal BCAA concentrations during wk6, wk7 and wk8 (see Lines 159-160 in the revised manuscript). Furthermore, adonis analysis

based on bray-curtis beta diversity measures showed, both week ($P < 0.001$) and treatment ($P = 0.001$) altered the microbiome configurations extensively (see Lines 188-194 in the revised manuscript). Based on the above mentioned, dietary changes did affect the microbial diversity and metabolites in the faeces.

Finally, in addition to dietary effect, three distinct phases of microbial progression were noted irrespective of dietary treatments as goats grew up. This is quite similar to the previously reported developmental phases in humans. That is what we wanted to talk in Fig. 6 and “Microbiome configurations and predicted functions revealed a temporal pattern of succession irrespective of dietary treatments” part, please see Lines 216 to 251 in the revised manuscript. If you have further inquiry or suggestion, please feel free to contact us.

3. Figure 5: the results are very interesting. Thus, how to link the results of 5B to three bacterial genera (Eubacterium, Faecalibacterium, and Bifidobacterium) in 5A? why not showing the progression of those three bacteria genera over time? It is crucial to determine their roles. In contrast, some other bacteria except for Bifidobacterium were shown in Figure 6C. It is confusing. Whether some metagenomic evidence can be provided to confirm the contributions of Eubacterium, Faecalibacterium, and Bifidobacterium to some distinct metabolic phenotypes. Here, it is unclear why only Eubacterium was highlighted in the graph in Figure 7 but exclude Faecalibacterium, and Bifidobacterium.

Reply: Thanks for your constructive comments. Good point! Firstly, sorry to make you confused. Those three bacteria genera in Fig.5A were landmark differential bacteria caused by treatments. Some landmark differential bacteria in Fig.6C were identified at different time periods in both groups. However, according to your suggestions, we have added Eubacterium and Faecalibacterium in Fig.6C to show the progression of those three bacteria genera over time. Thereafter, we added relevant information in the manuscript. Please see Lines 206-210, 229-234, 338-339 and Fig.6 in the revised version.

Secondly, in respect to why only *Eubacterium* was highlighted in the graph in Figure 7, please let us explain. We cannot agree with you more that further metagenomic evidence could assist in explaining the contribution of gut microbiota to distinct metabolic phenotypes. This validation work is in progress and more powerful data will be provided in the future. Despite these, in the current work, inferred microbial function and metabolite analysis based on SCFA profile both confirmed that HRUP treatment enhanced butyrate production. It is generally accepted that *Bifidobacterium* was associated with gut lactate and formate production, while not butyrate production. Meanwhile, although gut microbes belonging to *Eubacterium* and *Faecalibacterium* possessed the capacity to produce butyrate, only *Eubacterium* abundance was enhanced by HRUP treatment. Therefore, it is plausible to infer that the enrichment of *Eubacterium* by HRUP treatment contributed most to the metabolic phenotypes characterized by enhanced butyrate production; and only *Eubacterium* was highlighted in the graph in Figure 7. These information have been added in the revised manuscript, please see Lines 184-187.

If you have further inquiry, we are happy to answer.

4. Figure 6: it is arbitrary to divide time points into three phases because no obvious pattern was observed, for example (6A), for HRUP treatment, a clear decrease occurred in wk4 and reached to the levels of wk7 and wk8. Correspondingly, in Figure 6C, most of bacterial genera did not show similar patterns like those in Figure 6A. Hence, please refine those results.

Reply: We thank the referee for this valuable suggestion and apologize for the unclear description. Firstly, the three distinct phases were divided based on between-group (Fig.4B) and within-group (Fig.6A) bray curtis distance. On one hand, the pairwise PERANOMA analysis indicated that bacterial diversity in the early (wk0-2), middle (wk3-5) and late phase (wk 6-8) was different from each other ($P < 0.01$, Fig.4B). On the other hand, a wave crest appeared in within-group dissimilarity during middle phase, where the Bray-Curtis dissimilarity among different individuals

within each week was the highest, greater than those in early and late phases ($P < 0.05$, Fig.6A).

Secondly, Fig.6A showed the within-group bray curtis dissimilarity, while Fig. 6C showed progression of specific bacterial genera over week. In the revised Fig. 6C, five genera were prevalent in the early phase, while three genera dominated the late phase. Hence, they did not show similar patterns.

All those results were refined in the revised version. Please see Lines 218-226. If you have further suggestions, we are happy to follow.

5. Fig.7 showed charming models or pathways but related evidence needs to be further strengthened. Here, there is over-interpretation for microbial analysis.

Reply: We really appreciated for the reviewer's valuable and earnest comment. It is of particular importance to further validate and increased the power of our data through omics-based, culture-based or transplantation-based of intestinal microbiota. We cannot agree you with more that there is a little over-interpretation for microbial analysis. Hence, we preferred to keep Fig.7, while added some relevant information to avoid over-interpretation. Please see Lines 364-366 in the revised version. If you have any further inquiries or suggestions, please let us know.

March 16, 2022

Dr. Jinzhen Jiao
Institute of Subtropical Agriculture, CAS
Changsha, Hunan 410125
China

Re: Spectrum02545-21R1 (Enhancing metabolic efficiency through optimizing metabolizable protein profile in a time progressive manner with weaned goats as a model: involvement of gut microbiota)

Dear Dr. Jinzhen Jiao:

Your manuscript has been accepted, and I am forwarding it to the ASM Journals Department for publication. You will be notified when your proofs are ready to be viewed.

Sincerely,

Jinxin Liu
Editor, Microbiology Spectrum

Journals Department
Supplemental file 1: Accept